

# Single-cell RNA-seq analysis revealed the stemness of a specific cluster of B cells in acute lymphoblastic leukemia progression

Guifang Wang[1], Ensheng Zhang[2], An Chen[3] and Dachuan Meng[2]

[1] Department of Pediatric Medicine, The Second Affiliated Hospital of Shandong First Medical University, Tai'an, Shandong, China
[2] Department of Pediatric Hematology, Shandong Maternal and Child Health Hospital, Jinan, Shandong, China
[3] Department of Otolaryngology, The Second Affiliated Hospital of Shandong First Medical University, Tai'an, Shandong, China

Corresponding author
Dachuan Meng,
dachuan20010307@163.com

## ABSTRACT

**Background:** Childhood acute lymphoblastic leukemia (ALL) is a common pediatric cancer. The heterogeneous characterization of B cells in ALL progression poses new challenges to researchers. We used single-cell sequencing to explore the critical role of B cells in regulating the ALL immune microenvironment.

**Method:** We collected the single cell (sc) RNA-seq data of ALL and health sample from the gene expression omnibus (GEO) database, the "Seurat" and "harmony" R package was used for quality control and scRNA-seq analysis, in which the CellMarker2.0 database was used for cell type annotation. Subsequently, the FindAllMarkers function was used to identify the differentially expressed genes (DEGs) among various cell types and the DAVID database was applied for the biological process of DEGs. Then, the "inferCNV" package was used for copy number variation, regulons and cell communication were performed by SCENIC tool and CellChat package. The role of the target gene in regulating ALL progression was assessed using RT-qPCR, Transwell and scratch healing assays.

**Results:** We identified nine mainly cell clusters after scRNA-seq analysis, in which the B cells had higher infiltration proportion in the ALL samples and were sub-clustered into five cell sub-groups. The B cells 1 is closely associated with cell proliferation and stemness (TNFAIP3 and KDM5B), and the significant CNV of amplification occurred on chr6 and chr21 that supported stemness of B cells1. RXRB is a key transcription factor mediated the proliferation of B cells 1, which in turn suppressed hematopoietic stem cells (HSCs) proliferation and promoted cytotoxic NK/T cells activation through diverse cell communication ways. One of the key regulators of B cells is MYC, which promotes the migration and invasive ability of cell line leukemia cell lines.

**Conclusion:** This study reveals the stemness characteristics of B cells and their critical role in ALL progression, a finding that provides new potential directions for the development of targeted therapies against ALL.

## INTRODUCTION

Childhood acute lymphoblastic leukemia (ALL) is a common hematological pediatric cancer that account for approximately ~25% of all cancer cases and 75–80% of leukemia among younger than 15 years persons (*Malard & Mohty, 2020*), representing more than 50,000 new diagnosed cases each year globally (*Pui et al., 2018*). This disease is characterized by the failed maturation/differentiation and uncontrollable proliferation of bone marrow lymphoid precursors (such as T or B naive lymphocytes) that occurred sequential genetic alterations (*Terwilliger & Abdul-Hay, 2017*). The apoptosis and differentiation of these leukemia cells were inhibited, leading to they are abnormal proliferation in bone marrow and hematopoietic tissues and infiltrates distal tissues and organs affecting normal hematopoiesis of bone marrow (*Yan et al., 2021*). Currently, the cure rate of ALL is increasing and its 5-year survival rate is over 90% in developed countries due to the available of innovative protocol-driven treatments (*Pieters et al., 2016*), only 15–25% relapse after recovery caused the death of patients, so the childhood ALL is being as the most curable cancers (*Lamore et al., 2021*). But its long-term side consequence or late adverse effects of intensive chemotherapy and cranial radiotherapy (CRT), such as cardiometabolic complications, decreased fertility, neurocognitive dysfunction, endocrine disorder, pulmonary injury, thrombosis and treatment related second malignancy risk are not negligible during a vulnerable period of a child's development (*Barnea et al., 2015*), its cumulative incidence of chronic disease is over 60% (*Immonen et al., 2021*). Cohort studies of survivors after recover demonstrated that survivors frequently have diabetes, hypertension, osteonecrosis, attention and memory decline and depression that caused by the drug toxicity and radiation exposure (*Bhakta et al., 2017*). Therefore, mounting current research are mainly focused on reducing late adverse effects in comorbidities, improving survivors' quality of life and preventing these complications, suggesting that we are still lacking in understanding the etiological mechanism of ALL, particularly its recurrence, and developing more effective treatment methods.

Childhood ALL represent a complex disease involving multiple subtypes that occurred with distinctive somatic gene mutation, such as point mutations, chromosomal rearrangements, and aneuploidy (*Bloom et al., 2020*). These key genetic alterations are necessary to leukemogenesis through altering metabolism regulatory processes, blocking differentiation, subverting normal proliferation control and promoting resistance to apoptotic signals (*Hunger & Mullighan, 2015*). During the progression of ALL, interactions between immune cells play a crucial role in the onset and development of the disease. Immune cells communicate through mechanisms such as cytokine secretion, ligand-receptor binding, and signaling to regulate immune responses in the tumor microenvironment (*Pastorczak et al., 2021*). For example, immune cells such as B cells, T cells and NK cells are able to regulate the proliferation and survival of leukemia cells through complex interactions with hematopoietic stem cells (HSCs) (*Oberoi et al., 2020*; *Perez-Martinez et al., 2011*; *Glushkova et al., 2024*). Notably, based on the understanding of this molecular basis of ALL, accurate subtype stratification and cluster characteristics of

various immune cells may be providing several the novel insight development of innovative therapies through the next generation sequencing (NGS) technologies. Especially, the single-cell RNA-sequencing (scRNA-seq) technology is being a useful tool to elucidate the molecular distinction (simultaneously assessing both common and rare variants in genetic studies) of various cell type and enable cell profiling of tumor at single-cell resolution (*Shendure & Ji, 2008*), promoting the development of biomarker and personalized therapeutic strategies. Lin et al, reported three nicotinamide adenine dinucleotide (NAD) metabolism-related genes that are significant highly expressed in common myeloid progenitors (CMP), megakaryocyte-erythroid progenitor (MEP), and granulocyte-macrophage progenitor (GMP) cells supporting ALL relapse by using the scRNA-seq analyis (*Lin et al., 2022*). In addition, the single cell sequencing researches also provided several important insights into the cell lineage plasticity of ambiguous acute leukemia lineage (*Patel & Weinberg, 2020*), in which immunophenotypic classification of acute undifferentiated leukemia (lacking specific lineage differentiation type) and mixed-phenotype acute leukemia (markers > 1 lineage) remain controversial to appropriate therapeutic approach (*Alexander & Orgel, 2021*). More recently, studies found that they arise a common primitive hematopoietic progenitor (*Alexander et al., 2018*), another study demonstrated that the early T-cell precursor ALL and T/myeloid MPAL with similar immunophenotype were classified as separate entities (*Sin & Man, 2021*), but they originated from a hematopoietic stem/progenitor cells (HSPCs) undergoing diverse genomic rearrangements by using scRNA-seq analysis (*Di Giacomo et al., 2021*).

Here, we used single-cell RNA sequencing in order to analyze the immune cell composition in ALL samples. Bioinformatics-based tools were used for quality control and cell type annotation of scRNA-seq data, which in turn led to the identification of different cellular subpopulations and differentially expressed genes, and the analysis of functional enrichment. In addition, we used copy number variant analysis and transcription factor regulatory network analysis to explore the molecular signatures of specific cellular subpopulations. Finally, cellular experiments were performed in order to further confirm the role of key genes and regulatory networks in ALL progression. Our findings provide new perspectives for the understanding of the immune microenvironment in ALL, which in turn provides new directions for the diagnosis and treatment of patients with ALL.

## MATERIALS

### Data collection and processing

The publicly available scRNA-seq data originated from the GSE132509 dataset were downloaded from the gene expression omnibus (GEO, (https://www.ncbi.nlm.nih.gov/geo/)) database, including the two high hyper diploid (HHD) acute lymphoblastic leukemia and three healthy peripheral blood sample that sequenced on Illumina HiSeq 4000 (Human) (*Caron et al., 2020*). Subsequently, the "Seurat" R package was used for the scRNA-seq analysis, in which the Read10X function was used to read the expression matrix (*Zulibiya et al., 2023*). To obtain the high-quality scRNA-seq data, the filtering procedure to the raw matrix was performed for each cell including the cells contained 200–5,000 genes that expressed at least three cells with <10% mitochondrial gene ratio.
Next, the we used the SCTransform function to normalize the combined data and the "RunPCA" function was used for Principal Component Analysis (PCA), the harmony package was applied for the batch effect removing. Then, the UAMP reduction dimension analysis was performed by using the RunUMAP function to reduce the data complexity for cell clustering (*Narayan, Berger & Cho, 2021*), the FindNeighbors and FindClusters function was used for cell clustering (setting dims = 1:20, resolution = 0.1) (*Zulibiya et al., 2023*). In addition, above same process was performed for B cell subtype (dims = 1:20, resolution = 0.3). For cluster annotation, the gene markers provided by the CellMarker2.0 database was used for cell cluster annotation.

## Analysis of differentially expressed genes among cell sub-clusters

To explore the difference of gene expression pattern among various cell sub-clusters, the FindAllMarkers function was used to calculate the differentially expressed genes (DEGs) of each cluster by using the Wilcoxon-Mann-Whitney tests (setting logfc.threshold = 0.25, min.pct = 0.25, only.pos = TRUE) (*Zulibiya et al., 2023*). We up-loaded the interest genes to the Database for Annotation, Visualization and Integrated Discovery (DAVID, https://david.ncifcrf.gov/) database for the biological process enrichment analysis ($p < 0.05$).

## InferCNV (copy number variation) analysis

The infercnv R package was used to perform the CNV evaluation of each cell, and the CNVs of B cells were calculated and the Cytotoxic NK/T cells in healthy group were applied as the reference (setting parameters cluster_by_groups = TRUE, analysis_mode = "subclusters", (hidden Markov model) HMM_type = "i3", denoise = TRUE, HMM_report_by = "subcluster", HMM = TRUE) (*Lu et al., 2023*).

## SCENIC analysis for regulon activity assessment

The SCENIC algorithm is developed specifically for the gene regulatory networks (GRNs or termed as regulons) analysis upon transcription factors (TFs) and the regulons (TFs and its target genes) in individual cells. Based on the SCENIC official guidance, we used the GENIE3 method to calculate the potential targets of each TF, in which the motif dataset was used to build regulons for each TF followed by Spearman's correlation. Then, the AUCell function (Area Under the Curve) calculate the activity score of regulons in each cell based on the gene expression value, the higher score indicates the higher activated activity and the highly credible GRNs were presented by the cytoscape software (*Wang et al., 2023*).

## Cell-cell communication analysis

To investigate the potential interactions relationship among B cell sub-groups, hematopoietic stem cells and cytotoxic NK/T cells in the tumor microenvironment (TME). The cell-cell communication analysis was conducted by using the CellChat R package, which contained a publicly available repository of curated ligands and receptors for cell sub-groups interaction network (*Wang et al., 2023*). Then, enriched receptor-ligand interactions among various cell type were visualized by netVisual_bubble function based

on the receptor expression of one cell type and the corresponding ligand expressed by another cell type.

## Cell culture and transfection

Human bone marrow stromal cells HS-5 (BNCC339313) were purchased from BNCC (Beijing) Biotechnology Co. Ltd. and the chronic granulocytic leukemia cell line HAP1 (item #C631 bath 29663) was purchased from Horizon Discovery. Cells were cultured in Dulbecco's modified Eagle medium (11965; Gibco, Grand Island, NY, USA) and supplemented with 10% fetal bovine serum (26140-095; Gibco, Grand Island, NY, USA) and 1% antibiotics ((15070-063; Gibco, Grand Island, NY, USA). -092) cultured in Dulbecco's Modified Eagle medium (11965-092; Gibco, Grand Island, NY, USA) supplemented with 10% fetal bovine serum (26140-095; Gibco, Grand Island, NY, USA) and 1% antibiotics (15070-063; Gibco, Grand Island, NY, USA). Cells were cultured at 37 °C and 5% $CO_2$. Lipofectamine 2000 (Invitrogen, Waltham, MA, USA) was used to transfect cells with negative control (NC) and MYC siRNA (Sagon, China). The target sequence of MYC siRNA was 5′-AACTATGACCTCGACTACGAC-3′.

## QRT-PCR

RNA was reverse transcribed to cDNA by Qiagen One-Step RT-PCR kit (Qiagen Gmbh, Hilden, Germany) and subjected to qRT-PCR experiments. Amplification experiments were performed in an ABI 7500 system (Thermo Fisher, Waltham, MA, USA) using SYBR Green.

## Scratch healing assay

The cell line was inoculated in a 6-well plate, and when the adherent wall grew throughout the bottom, a scratch was made vertically by using a 200 μL pipette tip, rinsed twice with phosphate buffer, and photographed under an inverted microscope after 0 and 48 h of the scratch, respectively. Scratch healing rate = (0 h width–48 h width)/0 h width × 100%, test 3 times, re-well 2 times.

## Transwell analysis

A total of 30 μL of Matrigel was applied to the bottom of the upper chamber (serum-free medium) of the Transwell experimental setup (Corning, Corning, NY, USA), and cells were subsequently inoculated into the upper chamber, followed by the addition of PRMI-1640 medium containing 10% FBS to the lower chamber, and the non-migrated cells in the upper chamber were removed after incubation at 37 °C for 1 d. The cells were fixed with 4% paraformaldehyde and stained with crystal violet for 30 min. Cells were observed under a microscope and counted.

## Statistical analysis

All statistical analyses and visualization were performed by R software (version 4.3.2), an unpaired Wilcoxon rank-sum test and two-sided paired Student's t-test was used for difference analysis and $p < 0.05$ was regarded as statistical significance.

## RESULTS

### Highly proportion of B cells in the ALL samples after scRNA-seq profiles analysis

To investigate the cellular diversity in HHD acute lymphoblastic leukemia (ALL), we implemented a scRNA-seq analysis on two ALL and three heathy samples and obtained gene expression profiles of nine mainly cell subgroups after initial quality control described previously (Figs. S1A–S1D), including the hematopoietic stem cells (1, 2 and 3), B cells, myeloid cells, plasma B cells, plasmacytoid dendritic cells (pDCs), naïve T cells and cytotoxic NK/T cells (Fig. 1A), the top5 DEGs of each cellular sub-group were presented in bubble plot (Fig. 1B) and the expression patterns of the cluster-specific marker genes were demonstrated (Fig. 1C), such as the TCF7 was overexpressed in the Naïve T cells. The proportions of the cellular sub-groups diverse well among various samples, especially, the number of B cells in the healthy samples (Fig. 1C) are obvious less than that in the ALL groups (Fig. 1D), suggesting that the occurrence of ALL acute lymphoblastic leukemia is closely associated with the uncontrollable proliferation of B cells.

### B cells 1 cluster with stemness cell feature may be a key pathogenesis of ALL

Mounting studies reported that the childhood ALL is a common hematological malignancy characterized by uncontrollable proliferation of marrow lymphoid precursors, such as the pre-B cells, the huge alteration of B cells including the mature and pre-B cells in ALL samples were also observed in our sc-RNS-seq analysis results. Therefore, we further performed the cellular sub-clustering of these B cells and annotated five mainly B cell sub-groups from B cell (1) to B cell (5) (Fig. 2A). We analyzed the biological process (BP) of DEGs in each cell cluster by DAVID tool and found that the cell cycle, positive regulation of Wnt signaling, hematopoietic stem cell differentiation and B cell proliferation pathways were significantly enriched in the B cells 1 group supporting B cell proliferation (Fig. 2B), while a number of immune-related pathways, such as positive regulation of T cell differentiation and activation, leukocyte activation and inflammatory response pathways were enriched in others B cell clusters (Figs. 2C–2F). We examined the subtype-specific-marker genes and identified that the markers of cancer stemness feature including the TNFAIP3 and KDM5B were also expressed in the B cell 1 and the markers of plasma cells were expressed in the B cells 3 (Fig. 2G), suggesting that the B cells 1may be the stemness B cells and the B cells 3 may be the mature plasma B cells. The proportions of these B cell clusters showed that the B cells 1 is a specific pre-B cells that only existed in the ALL samples with larger cell number (Fig. 2H), indicating that the B cells 1 with stemness feature may a key underlying pathogenesis of ALL.

### Chromosome amplification in B cell 1 may support its stemness properties

To distinguish the stemness feature of B cells 1, we calculated and identified the copy number variation (CNV) on 22 pair euchromosome by using the inferCNV for each B cells
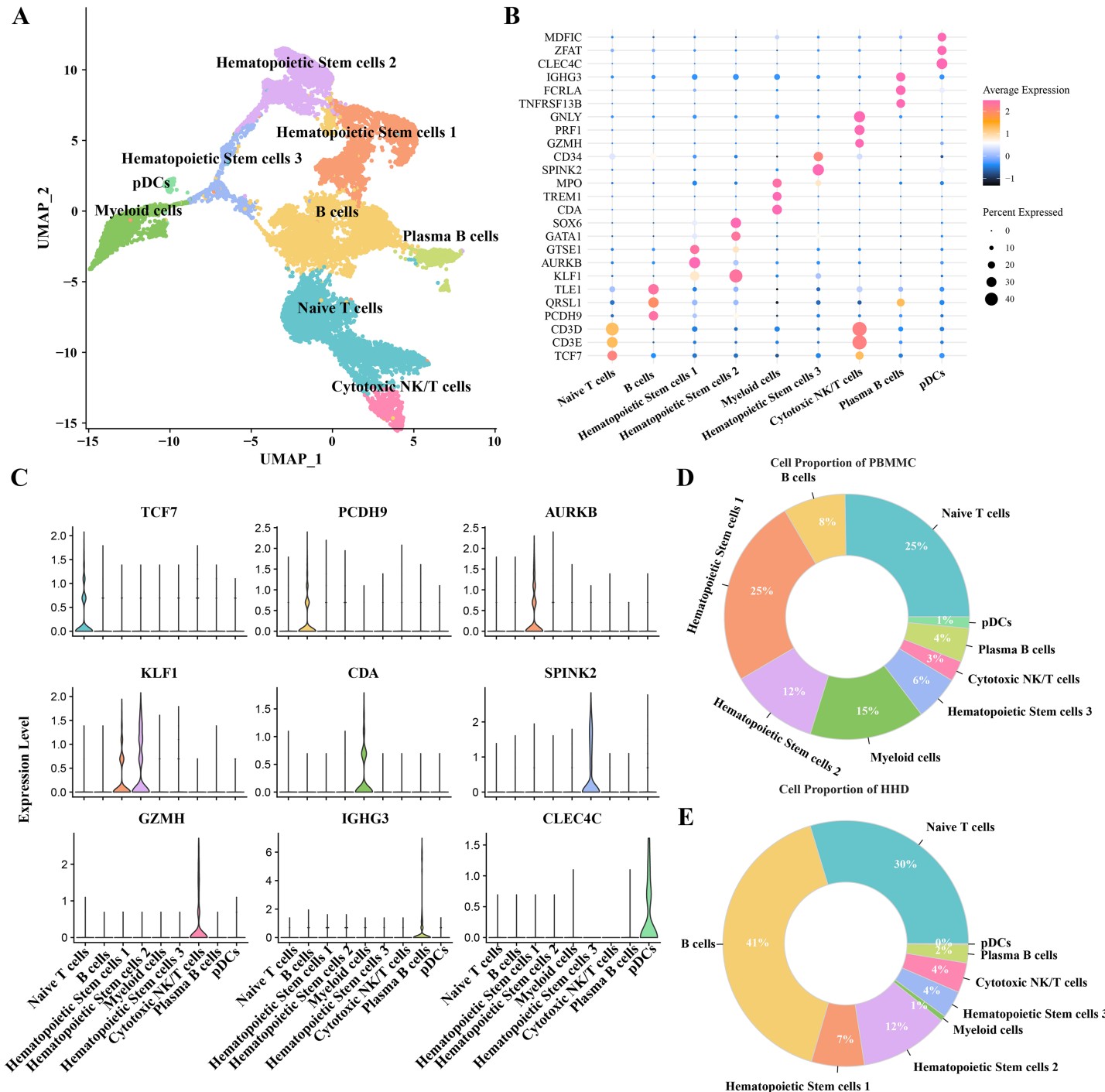

**Figure 1** **The scRNA-seq landscape of pre-B high hyper diploid ALL.** (A) UMAP plot of nine mainly cell clusters. (B) The bubble plot of marker genes expression in each cell subtype. (C) The specific-marker genes of each cell. (D) The cell proportion analysis in the healthy samples. (E) The cell proportion analysis in the ALL samples.

1 according to the transcriptomes. An inferCNV clustered heatmap was generated, the normalized expression of Cytotoxic NK/T cells were plotted in top panel and the B cells 1 are in bottom panel. Meanwhile, the gain regions are depicted in red and the loss regions

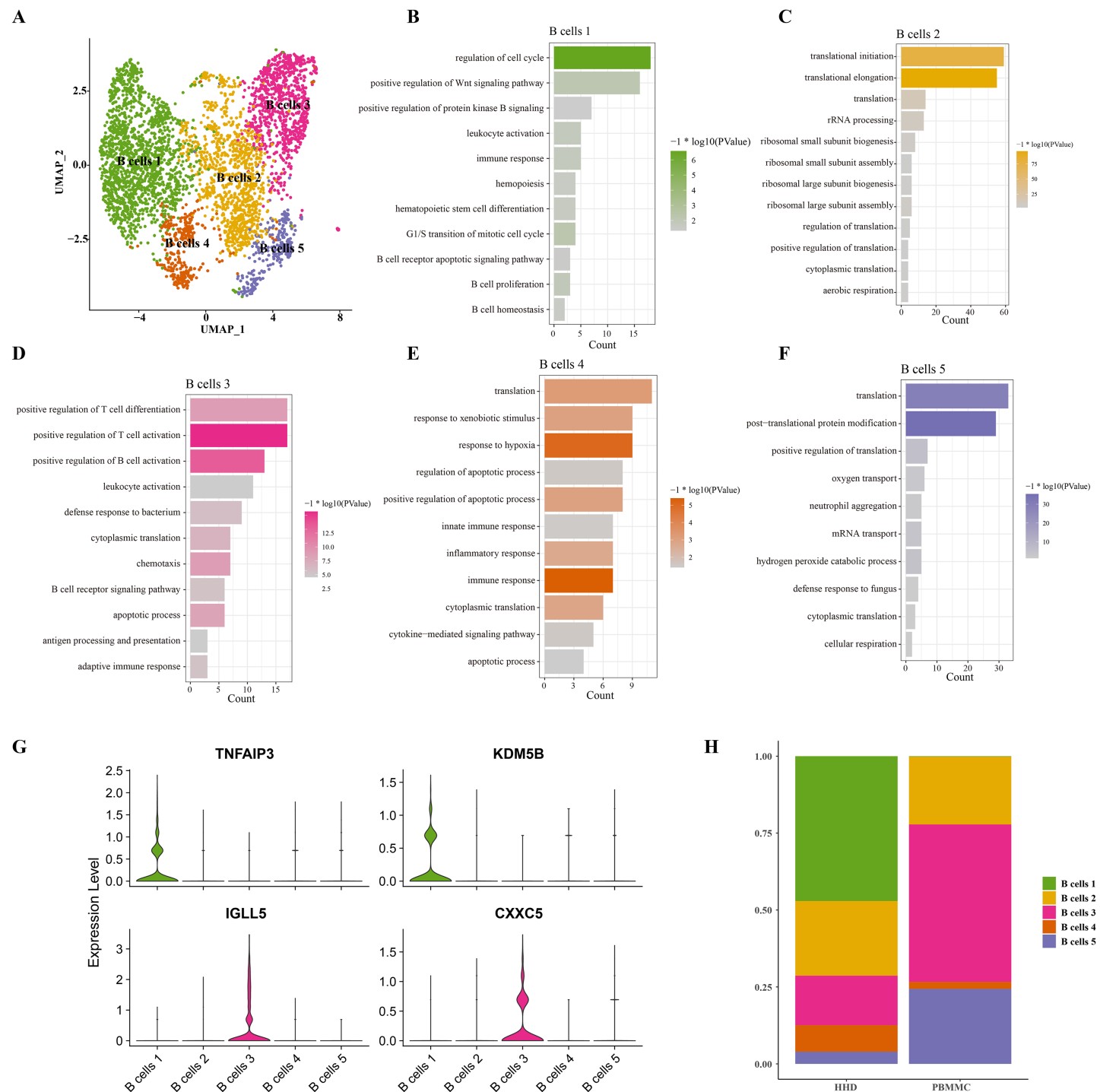

**Figure 2 The scRNA-seq landscape of B cell sub-groups.** (A) UMAP plot of five mainly B cell clusters. (B) The biological process of significant DEGs in the B cells 1. (C) The biological process of significant DEGs in the B cells 2. (D) The biological process of significant DEGs in the B cells 3. (E) The biological process of significant DEGs in the B cells 4. (F) The biological process of significant DEGs in the B cells 5. (G) The specific-marker gene expression of B cells 1 and B cells 3. (H) The cell proportion analysis of B cells in the healthy and ALL samples.

are in blue in the CNV heatmap. Results showed that the B cells 1 had relatively highly CNV levels compared with the Cytotoxic NK/T cells, we discovered that all B cells 1 experienced remarkable amplification events on chr6 and chr21 (Fig. 3A), implying that these regions may be confer the malignant stemness feature to B cells 1. Function enrichment analysis of these gain genes on chr6 and chr21 showed that the chr6 amplification genes are associated with the chromatin remodeling and organization, cell division and cycle (Fig. 3B), while the chr21 amplification genes are mainly associated with the type III interferon signaling pathway and regulation of JAK-STAT cascade (Fig. 3C). These pathways, designed for immune regulation and cell survival, also further support the malignant character of B cell 1 and its potential role in ALL progression.

### The key TF mediated the proliferation and stemness of B cells 1

To further reveal the underlying regulatory mechanism of Stem B cells 1, we used the SCENIC tool to analyze the TFs regulatory network in Stemness B cells 1. A number of proliferation-related TFs, such as MYC and STAT1 were identified (Fig. 4A), in which the regulon of MYC contained nine target genes (Fig. 4B) and the regulon of STAT1 included 17 target genes (Fig. 4C), suggesting the proliferation transcriptional regulation are essential to stemness of B cells 1. Further, the gene overlap analysis between regulons and chr6 amplification genes was performed, the key TF of RXRB was obtained (Fig. 4D), which affected 10 target genes expression in the GRNs (Fig. 4E).

### Transcription factors of B cells 1 regulate leukemia cells affecting disease progression

In this study, we explored the relative expression levels of key transcription factors in leukemia cells by RT-qPCR, and the results showed that MYC, RXRB, were significantly up-regulated, while STAT1 expression was down-regulated in HAP1 compared to HS-5 (Figs. 5A–5C). Since previous analyses demonstrated that MYC is one of the key molecules regulating stemness in B cells 1 cells, follow-up experiments on this molecule were performed in this study. Scratch healing experiments demonstrated that silencing of MYC resulted in downregulation of the migratory capacity of leukemia cell lines, implying a facilitating effect of MYC on the migratory capacity of leukemia cell lines (Figs. 5D and 5E). In contrast, Transwell experiments revealed the tangible promotion of MYC on the invasion ability of leukemia cell lines (Figs. 5F and 5G). All of the above findings illustrate the tangible regulatory role of MYC, a transcription factor of B cells 1, in leukemia progression.

### Stemness B cells 1 suppressed the cell activity of hematopoietic stem cells in ALL

Hematopoietic stem cells abnormal usually is regarded as a primary culprit of ALL and the B cells 1 was an identified abnormal proliferative cell with stemness in ALL samples. We utilized the cell-cell communication analysis to explore the relationship between stemness B cells 1 and hematopoietic stem cells. We found the stemness B cells affected the survival of hematopoietic stem cells *via* btla-tnfrsf14 (Fig. 6A) that mediated apoptosis (*Zhu & Lu,*

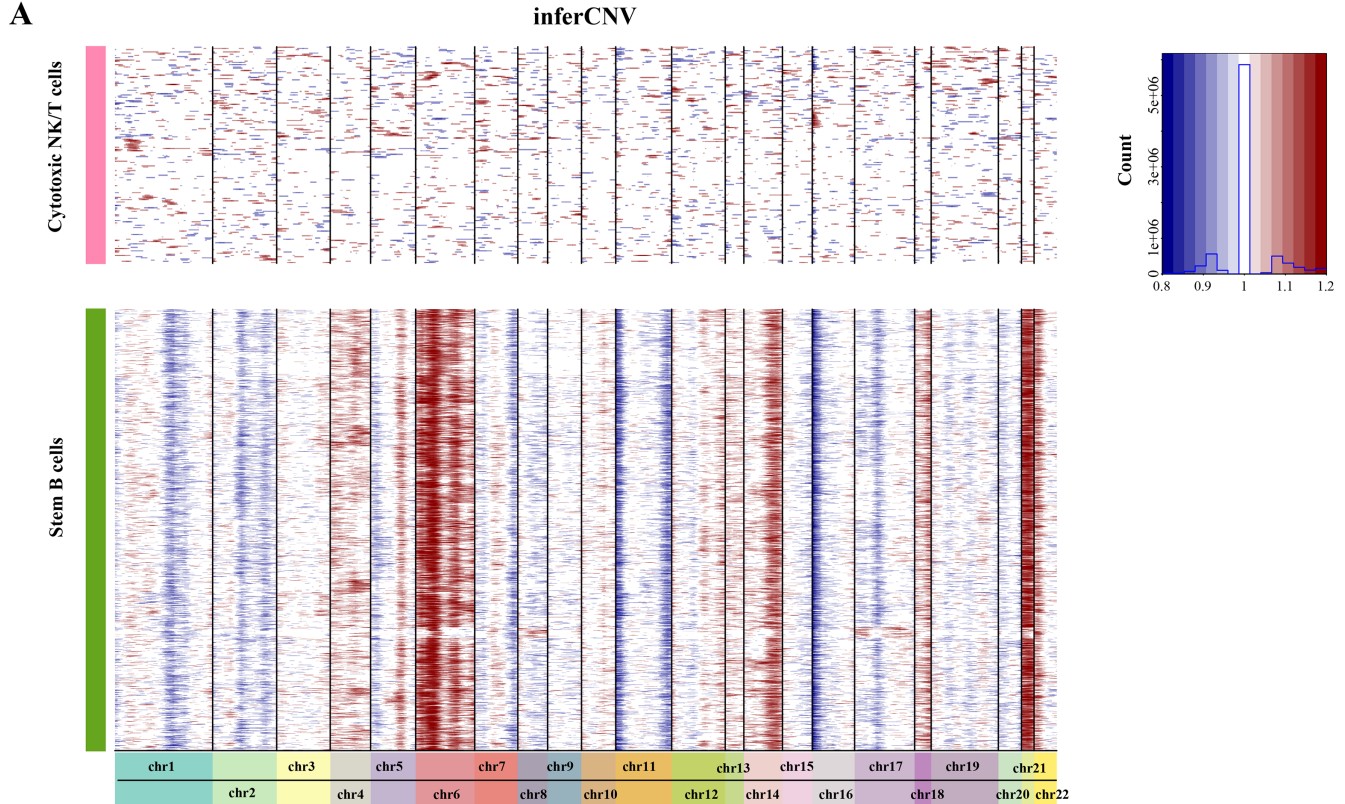

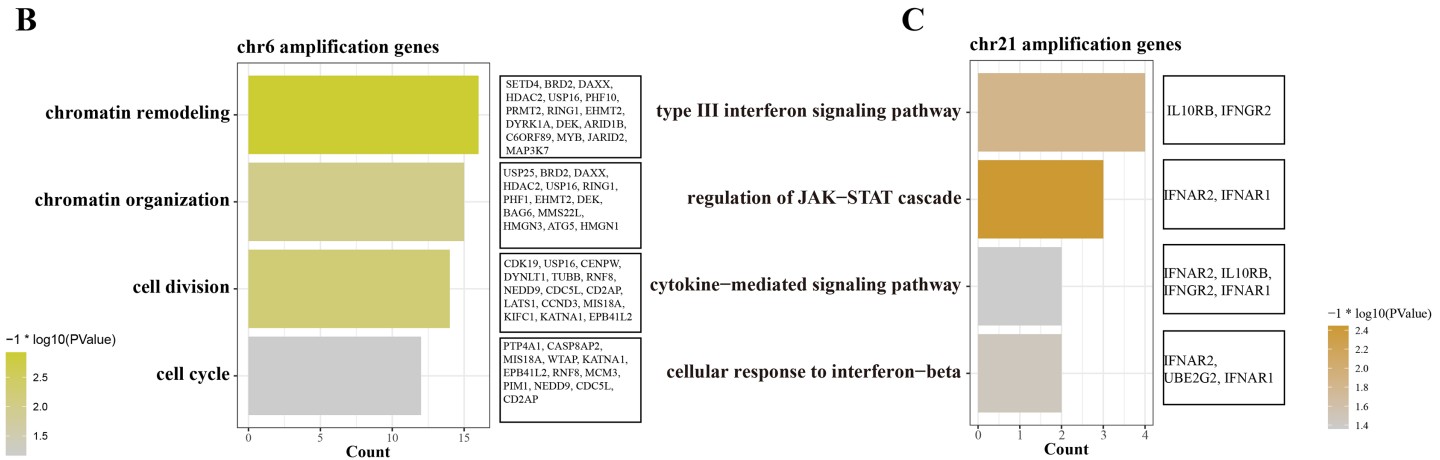

**Figure 3** **The landscape of CNC of stemness B cells 1.** (A) The heatmap of large-scale CNVs of stemness B cells 1 and cytotoxic NK/T cells. (B) The function enrichment analysis of chr6 amplification genes. (C) The function enrichment analysis of chr21 amplification genes.

*2018*) and inhibited cell proliferation (*Cheng et al., 2022*). MK was identified as a promoting differentiation factor of cancer at embryo stage (*Sorrelle, Dominguez & Brekken, 2017*), but the overexpressed MDK in the stemness B cells 1 may be suppress the proliferation of hematopoietic stem cells through binding the NCL receptor (*Wu et al., 2020*; Fig. 6B). In addition, the cytokines of LAT involved in the immune defense, cell

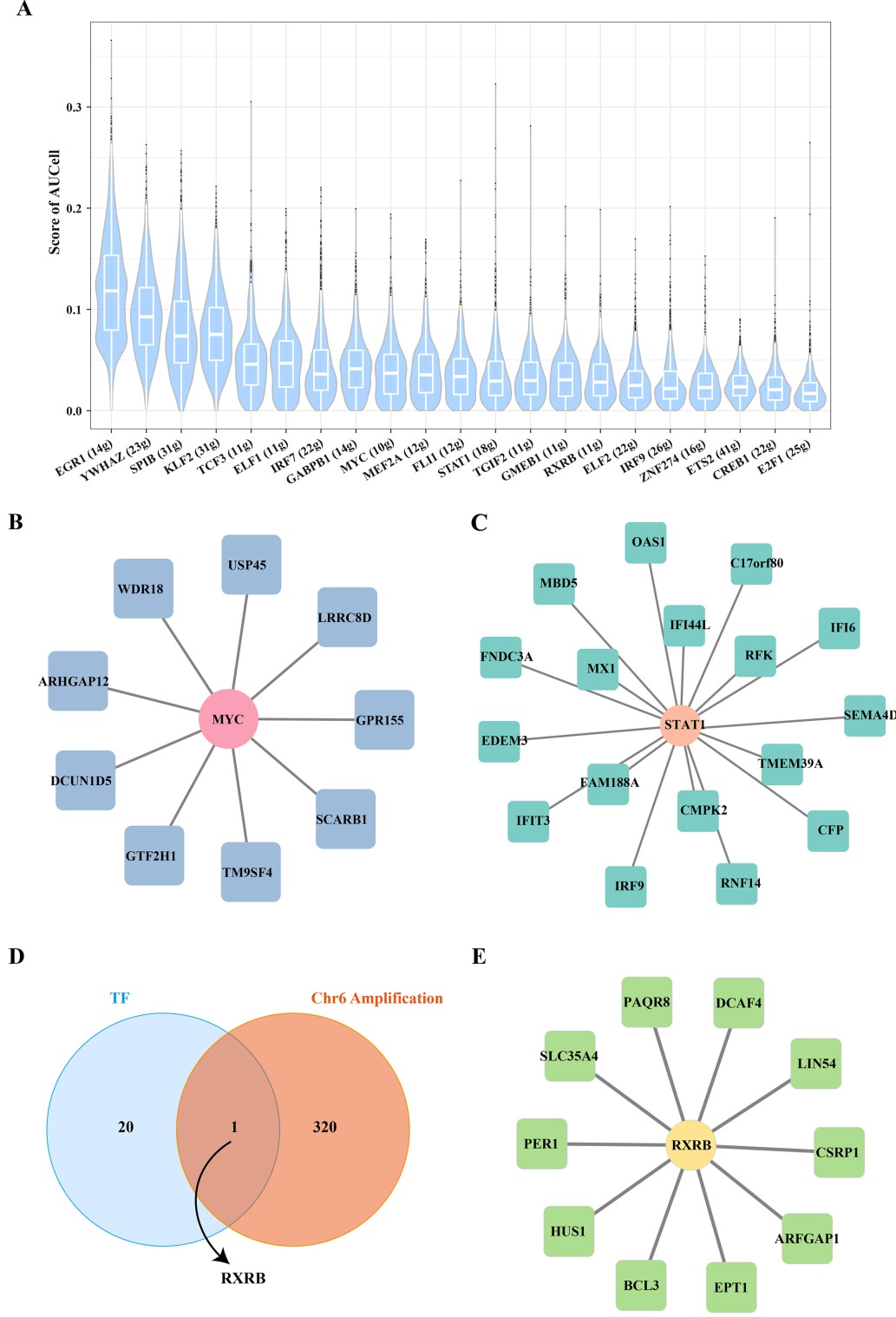

**Figure 4 Transcription factor regulatory network in stemness B cells 1.** (A) Box plot of AUCell scores for regulon in stemness B cells 1. (B) Regulatory network of MYC and cell proliferation-related target genes. (C) Regulatory network of STAT1 and cell proliferation-related target genes (the circles represent transcription factors and squares represent target genes). (D) Venn map of transcription factors and amplified genes of chromosome 6 in stemness B cells 1. (E) Regulatory network of RXRB and cell proliferation-related target genes.

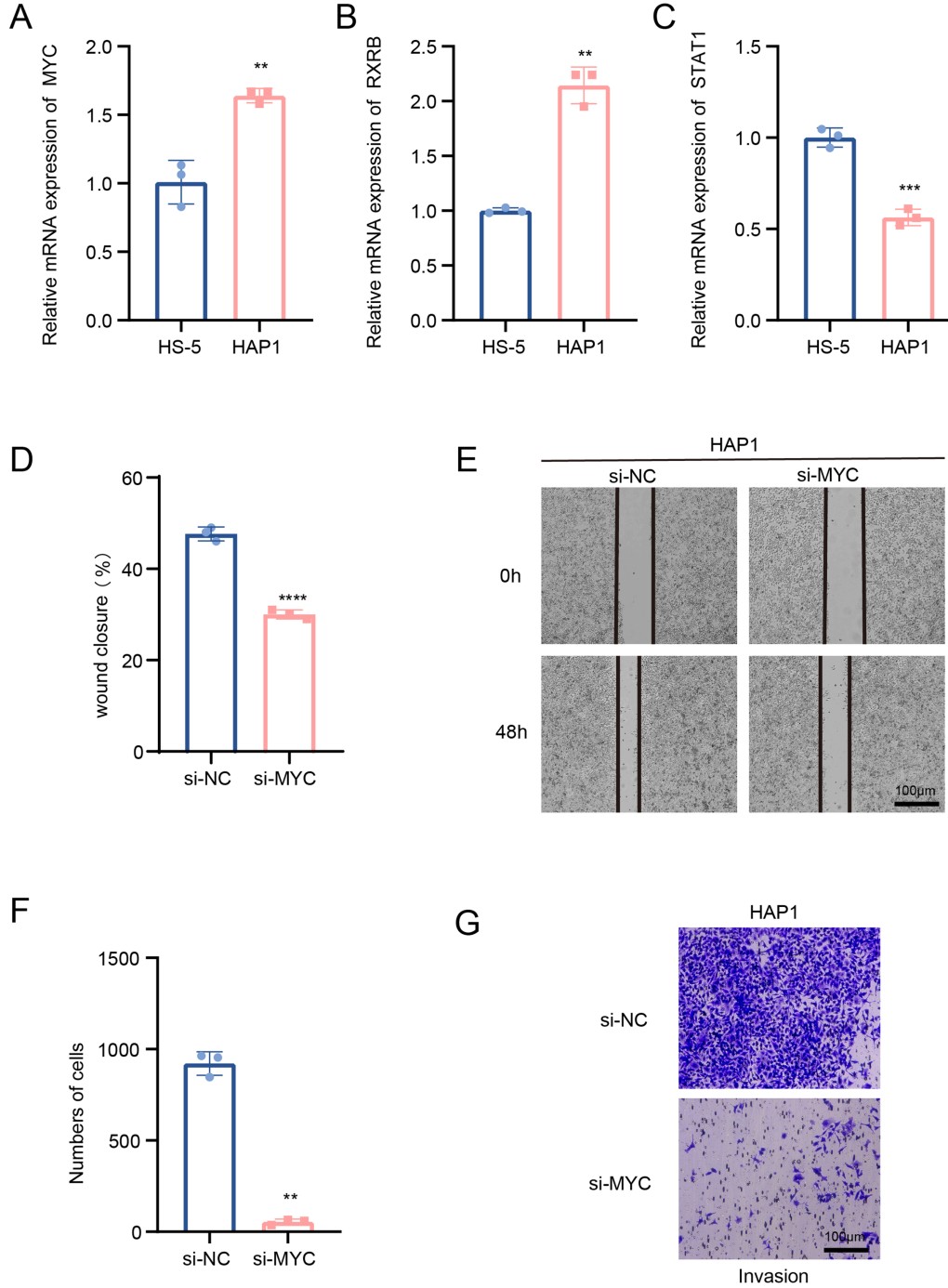

**Figure 5 Transcription factors of B cells 1 regulate leukemia cell lines.** (A) mRNA expression levels of MYC in HAP1 compared to HS-5. (B) mRNA expression levels of RXRB in HAP1 compared to HS-5. (C) mRNA expression levels of STAT1 in HAP1 compared to HS-5. (D) Relative scratch healing rate of MYC-silenced leukemia cell lines. (E) Scratch healing results of MYC-silenced leukemia cell lines at 0 and 48 h. (F) Quantification of invasion level of MYC-silenced leukemia cells in Transwell assay. (G) Transwell assay results of MYC-silenced leukemia cell lines. **$p < 0.01$, ***$p < 0.001$, ****$p < 0.0001$.

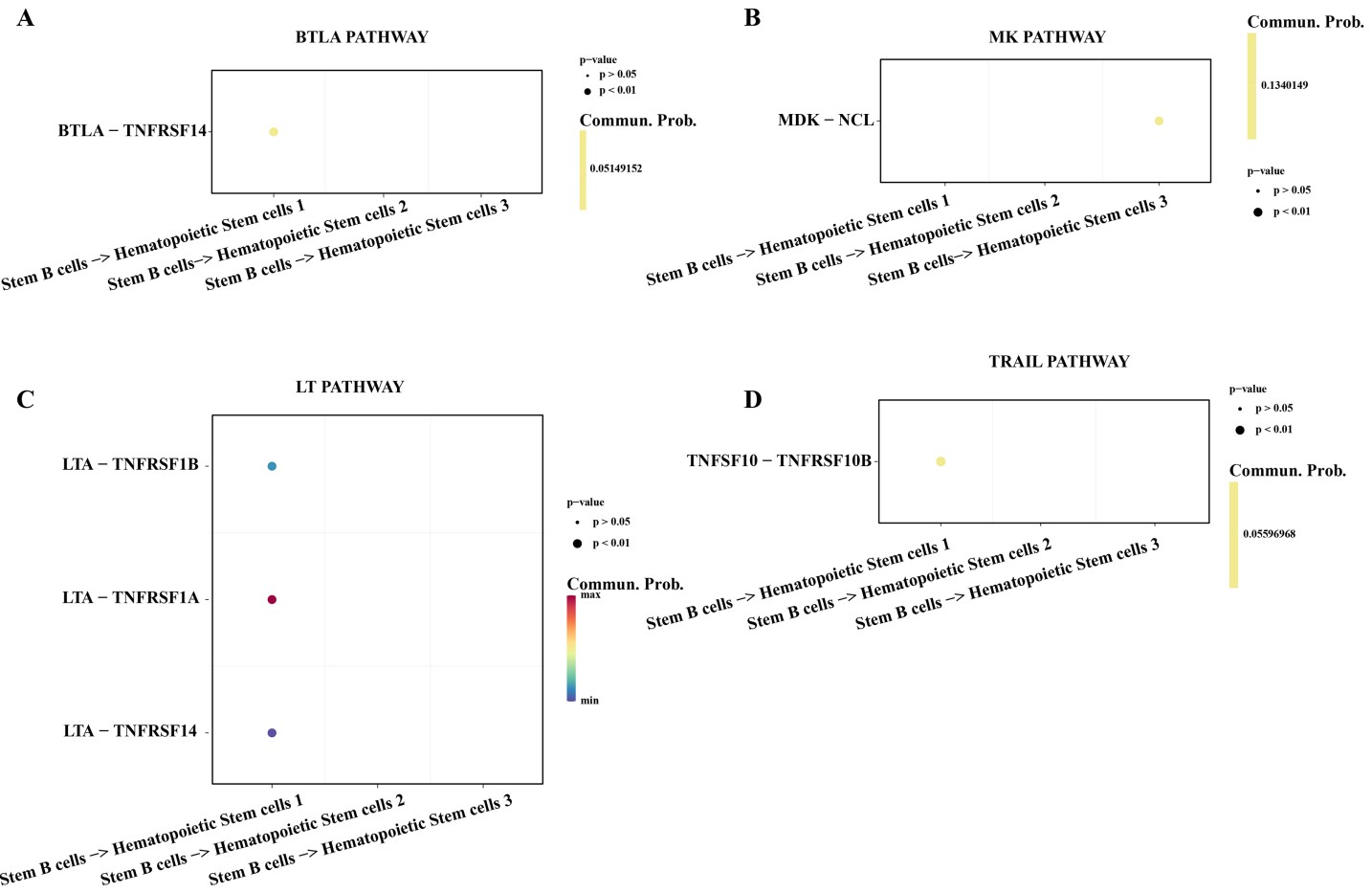

**Figure 6  Cell communication between the stemness B cells 1 and hematopoietic stem cells.** (A) Bubble plot of ligand-receptor pairs in the BTLA pathway. (B) Bubble plot of ligand-receptor pairs in the MK pathway. (C) Bubble plot of ligand-receptor pairs in the LT pathway. (D) Bubble plot of ligand-receptor pairs in the TRAIL pathway.                                                

proliferation and apoptosis for host homeostasis, the overexpressed TNFRSF14 in the hematopoietic stem cells can be induced by the LTA in B cells 1 for apoptosis (*Zhu & Lu, 2018*; Fig. 6C), and the overexpressed TNFSF10-TNFRSF10A pair may be also mediated the apoptosis of hematopoietic stem cells (*Zhao, Liu & Su, 2014*; Fig. 6D), suggesting that the stemness B cells 1 inhibit the hematopoietic stem cells survival.

## Stemness B cells 1 activated the activity of cytotoxic NK/T cells in ALL

Subsequently, we explored the affecting of stemness B cells 1 to cytotoxic NK/T cells. The results showed that a series of promoting proliferation signaling pathway and receptor-ligand pairs were identified. The overexpressed HLA-F in the B cells 1 can bind the receptor of CD8A in cytotoxic NK/T cells promoting the killer role (Fig. 7A), the chemokines CCL can bind the CCR5 receptor to induce T lymphocytes migration for anti-infection (*Sharapova et al., 2018*; Fig. 7B), ICOS can induce the activation, proliferation and survival of T cells through binding CD28 (*Blotta et al., 1996*; Fig. 7C), the TNFSF14 can promote the proliferation and effector of T cells (*Tamada et al., 2000*;

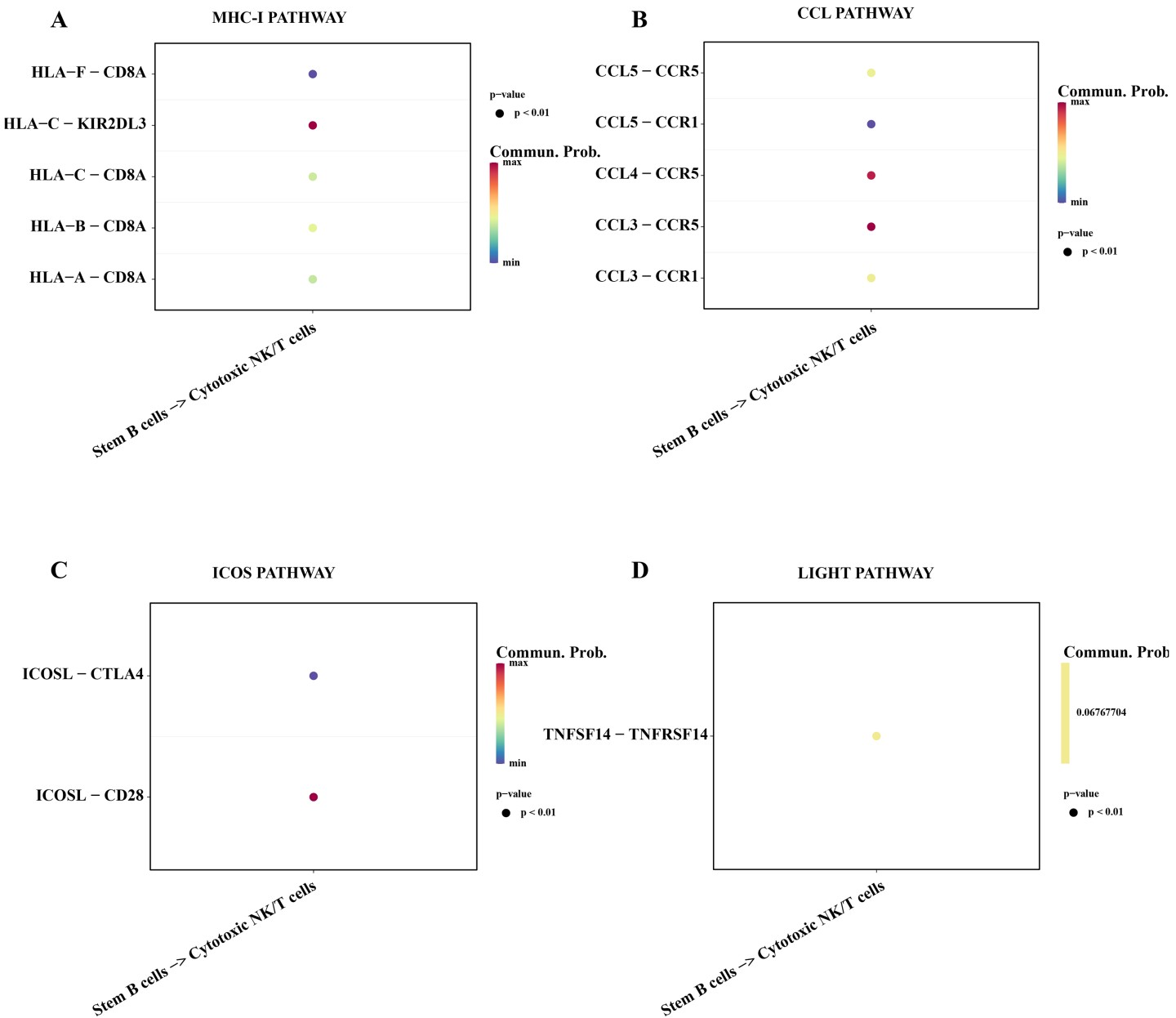

**Figure 7  Cell communication between the stemness B cells 1 and cytotoxic NK/T Cells.** (A) Bubble plot of ligand-receptor pairs in the MHC-I pathway. (B) Bubble plot of ligand-receptor pairs in the CCL pathway. (C) Bubble plot of ligand-receptor pairs in the ICOS pathway. (D) Bubble plot of ligand-receptor pairs in the LIGHT pathway.           

Fig. 7D), these results supported that the stemness B cells 1 promote the activity of cytotoxic NK/T cells.

## DISCUSSION

ALL is a common hematological pediatric malignancy characterized by the impaired differentiation and abnormal proliferation of lymphoid lineage cells. In recent years, the development of hematopoietic stem cell transplantation techniques and novel chemotherapeutic agents has significantly improved disease-free survival and remission

rates in patients with ALL (*Pui et al., 2010*); however, there are still some cases that are difficult to resolve after relapse that clone or die from recurrence of serious complications. So, exploring the underlying pathogenesis, drug resistance and chemotherapy insensitivity of leukemia cells supported the development of fewer side effects (reducing toxicities) and more effective drugs and therapeutic regimens are necessary and urgent (*Xu et al., 2022*; *Tian et al., 2022*; *Seyfinejad & Jouyban, 2022*). In this study, we performed a scRNA-seq analysis of the high hyper diploid (HHD) ALL and normal healthy samples, the results showed that the highly infiltration of B cells in the ALL samples are heterogeneous and sub-divided into five cell sub-groups, in which the B cells 1 exhibited stemness as a major contributor of ALL progression. Then, the CNV and function enrichment analysis revealed that the amplification genes on chr6 are closely associated with the cell proliferation and division process (such as CDK19, USP16 and CDC51), the transcription factor RXRB play a crucial role in mediating B cells 1 proliferation. In addition, we found that the B cells 1 affected the proliferation and apoptosis of hematopoietic stem cells through the MDK-NCL and LTA-TNFRSF14 recognition, while promoted the cytotoxic NK/T cells activity *via* the HLA-F-CD8A, CCL-CCR5 and TNFSF14-LIGHT binding.

B-lineage acute lymphoblastic leukemia (B-ALL) is one of the commonest subtypes of leukemia and represent ~85% of cases (*Linet et al., 2016*), and its prognosis is largely determined by the presence or absence of chromosomal rearrangements or gross aneuploidy (*Woo, Alberti & Tirado, 2014*). Such as the HHD-ALL cases with more than chromosomes had the most common t (12; 21) (ETV6/RUNX1) rearrangement, occurring in 20–25% of patients and representing nearly half of the encountered chromosomal anomalies in B-ALL and associating with a favorable prognosis (*Moorman et al., 2010*). In general, the normal B lymphocytes originated from the hematopoietic stem cells and matured through a series of developmental stages in bone marrow expressing specific cell surface markers and transcription factors (*Liu et al., 2019*), while most of B-cell ALL derived from clones that are stunted at pro-B cell or pre-pro-B cell stages that lacks prognostic or therapeutic relevance (*Campos-Sanchez et al., 2011*). ETV6-RUNX1 fusion usually is an initiating event *in utero* contributing to a preleukaemic state, but it is not sufficient for leukaemogenesis and the gene alterations of cell cycle and B-cell lineage differentiation, these genes such as PAX5, CDKN2A and CD5L, or chromosomal gains or losses also are essential for leukaemic transformation (*Greaves, 2018*). We identified the B cells 1 significantly expressed the TNFAIP3 and KDM5B markers, in which the tumor necrosis factor alpha inducible protein three (TNFAIP3) belongs to diverse deubiquitinase (DUBs) family and encode ubiquitin-modifying enzyme A20 (*Ciccacci et al., 2019*), which plays a crucial role in hematopoietic stem cells, the most persuasive evidence is that the TNFAIP3 defective animals died from acute systemic inflammation (*Nakagawa, Davis & Rathinam, 2018*; *Nakagawa et al., 2015*). KDM5B encode a jmjc domain-containing histone demethylase that was up-regulated in various cancer to represses the transcriptional function as an oncogene (*Li et al., 2020*), *Wang et al. (2024)*, revealed that upregulated KDM5B can enhance tumor malignancy, including the drug resistance and cancer cells stemness. These two genes as marker of B cells 1 enhance the stemness of B cells 1 and promote the malignancy grade of ALL, supporting the B cells 1 is a very crucial

contributor for ALL progression. In addition, the chr6 amplification genes are associated with the chromatin remodeling, cell cycle, division and proliferation further supported the basis of stemness B cells 1.

The retinoid X receptor beta (RXRB) is an identified transcription factor that encode retinoid X receptor and is widely expressed and play an important role in spermatogenesis (*Vernet et al., 2008*). The mutual heterodimers of RXRB and vitamin D receptor (VDR) promoted the development of cancer, and activated the Ras-Raf-MAPK-ERK signaling pathway to involve in the dominating VDR pathway for various gene expressions regulation (*Deeb, Trump & Johnson, 2007*). Another study reported that the RXRB acted as a downstream effector of RAB39A to foster cancer cell stemness (*Chano et al., 2018*), and interfered with the histone deacetylase (HDAC) to maintain embryonic neuronal stem cells (*Perissi et al., 2010*). We found that the stemness B cells 1 inhibited the proliferation and promoted the apoptosis of HSCs, the activity of HSCs was tightly constrained by the inflammatory stimuli and cholinergic signaling (*Fujii et al., 2017*), *Patel & Pietras (2022)* reported that the B cells located in the bone marrow produce the acetylcholine and prompts the release of cytokines from BM stromal cells to inhibit the blood-forming activity of HSCs against cardiovascular dysfunction, the phenomenon of stemness B cells 1 inhibiting the blood-forming activity of HSCs in ALL is value topic for further exploring. Meanwhile, the stemness B cells 1 activated the activity of Cytotoxic NK/T cells, which contributed to the pathogens eliminating and inflammation supporting. Cognate B cells and T cells interaction can lead to the B cell proliferation in extrafollicular (EF) with no affinity maturation trait (*MacLennan et al., 2003*) and these extrafollicular (EF) B cells in turn can accelerate the exhaustions of Cytotoxic NK/T cells (*Ma et al., 2024*), this may be a specific way of stemness B cells 1 promoting ALL progression. However, there are some limitations to our study. Although this study reveals the stemness characteristics of B cell 1 and its associated gene regulatory network, in-depth experimental validation is lacking at the level of specific molecular mechanisms. In addition, the small sample size limits the extensiveness and reproducibility of the findings. In the future, we will verify the generalizability of the study results by adding more patient samples, especially those from patients with different subtypes of ALL.

## CONCLUSION

The scRNA-seq data of peripheral blood samples supports the identification of distinct cell types and provide some novel perspective on pathogenesis of ALL. Results of clustering analysis, we characterized the heterogeneous of ALL and identified nine mainly cell subtypes (hematopoietic stem cells (1, 2 and 3), B cells, myeloid cells, plasma B cells, plasmacytoid dendritic cells (pDCs), naïve T cells and cytotoxic NK/T cells). The stemness feature of B cells 1 were analyzed by using the secondary clustering, copy number variation (CNV) and functional enrichment analysis of DEGs. In addition, their key target regulon and cell communication ways were also identified, which may be help to examine the stemness phenotype pathogenicity of B cells 1 in ALL. Overall, our findings can provide several new insights to understand the underlying pathogenic mechanism of ALL.

## ABBREVIATIONS

| | |
|---|---|
| **ALL** | Acute lymphoblastic leukemia |
| **CNV** | Copy number variation |
| **scRNA-seq** | Single cell RNA-seq |
| **GEO** | Gene Expression Omnibus |
| **PCA** | Principal Component Analysis |
| **BP** | Biological process |
| **CRT** | Cranial radiotherapy |
| **NGS** | Next generation sequencing |
| **DEGs** | Differentially expressed genes |
| **GRNs** | Gene regulatory network |
| **TFs** | Transcription factors |
| **NAD** | Nicotinamide adenine dinucleotide |
| **TME** | Tumor microenvironment |
| **HHD** | High hyper diploid |
| **EF** | Extrafollicular |
| **HDAC** | Histone deacetylase |
| **HSCs** | Hematopoietic Stem Cells |

### Funding

The authors received no funding for this work.

### Competing Interests

The authors declare that they have no competing interests.

### Author Contributions

- Guifang Wang conceived and designed the experiments, analyzed the data, prepared figures and/or tables, and approved the final draft.
- Ensheng Zhang conceived and designed the experiments, authored or reviewed drafts of the article, and approved the final draft.
- An Chen performed the experiments, authored or reviewed drafts of the article, and approved the final draft.
- Dachuan Meng performed the experiments, analyzed the data, prepared figures and/or tables, and approved the final draft.

### Data Availability

The datasets generated and/or analyzed are available at GEO: GSE132509.

https://www.ncbi.nlm.nih.gov/geo/query/acc.cgi?acc=GSE132509.

The raw data is available at GitHub and Zenodo:

- https://github.com/1Mdc/Updated-raw-data.git.

- 1Mdc. (2024). 1Mdc/Updated-raw-data: Updated raw data (v.1.1.1). Zenodo. https://doi.org/10.5281/zenodo.12650831.

## Supplemental Information

Supplemental information for this article can be found online at http://dx.doi.org/10.7717/peerj.18296#supplemental-information.

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
