# Peer review of "Single-cell RNA-seq analysis revealed the stemness of a specific cluster of B cells in acute lymphoblastic leukemia progression"

_PeerJ, doi:10.7717/peerj.18296_

## Round 0.1 · original submission · Major Revisions

We have now received comments from two expert reviewers, and I have carefully considered their feedback along with my own evaluation of your work. Both reviewers have highlighted the potential significance and novelty of your study. However, they have also raised several important concerns that need to be addressed before the manuscript can be considered for publication. Therefore, I invite you to submit a major revision of your manuscript. In your revision, please address all the points raised by the reviewers in detail. Please provide a point-by-point response to all reviewer comments along with your revised manuscript. Clearly highlight all changes made in the revised version. We look forward to receiving your revised manuscript within required timeframe.

Reviewer 1 ·

Basic reporting

In this study, the author explores the potential link between the stemness of a specific cluster of B cells and the development and prognosis of acute lymphoblastic leukemia. Prognostic factors and mutational signatures in acute lymphoblastic leukemia were identified through bioinformatics methods. The experimental design is rigorous. However, there are still some deficiencies in details in the manuscript.
1. In the conclusion of abstract, what significance do the results of this paper reveal.
2. Line 218, the amplification events on chr21 were associated with the type c interferon signaling pathway and regulation of JAK-STAT cascade. In what ways does it affect the stemness of B cells. Whether this subtitle is reasonable or not.
3. The results of TF network are important. Please gave a proper discussion.
4. What is the role of Cytotoxic NK/T cells in ALL progression, and has it been reported.
5. What are the prospects and shortcomings of this study.

Experimental design

6. Line 55-56, ALL was characterized by uncontrollable proliferation of bone marrow lymphoid precursors (such as T or B naive lymphocytes). In this study, the B cell has the largest proportion change in the HHD and control groups. How do we collect data if we want to study a sample with large T cell changes.
7. What is the relationship between stemness B cell and Hematopoietic Stem Cells. Is competition or derivative.
8. Stemness B cells has high plasticity, what factors can affect the polarization of this type cell.
9. Stemness B cells promoted the activity of Cytotoxic NK/T cells. Is this detrimental to disease progression.
10. What are the transcription factors that affect the high expression of TNFAIP3 and KDM5B.

Validity of the findings

no comment

Additional comments

no comment

Reviewer 2 ·

Basic reporting

The main thrust of this study was to investigate the infiltration type of immune cell subpopulations in acute lymphoblastic leukemia (ALL) by single-cell transcriptome analysis. This study first obtained sample data from public databases for ALL and clustered cell subpopulations by single-cell analysis. Follow-up studies explored the links between immune cell subpopulations in ALL and disease progression, metabolic regulation, and immune modulation, and validated some of the conclusions with joint cellular experiments. This is a bioinformatics joint cellular experiments study that meets the publication requirements overall, but the following issues still need to be addressed before publication:
1. The purpose of this study was to explore the mechanisms of immune cell regulation in ALL, so why didn't the analysis section focus on the link between immune cell marker genes and the level of infiltration of other cell subpopulations? Please provide an explanatory note in the methods section.
2. Are the immune cell-associated marker genes mined by this study addressed in the existing literature? Because the cellular experiments conducted in this study were too routine, it is recommended that the findings of this study be elucidated in depth in the context of the existing literature.
3. The abstract section of this study is not very general and does not highlight well the usefulness of bioinformatics tools for this paper, please elaborate on this.
4. The introductory part of this study does not clearly spell out what the intention of this study is, i.e., it does not state the necessity of studying the immune cell subpopulations of ALL, which leads to the idea of this paper being a tiger's tail, so please state in the introductory part the general idea of this study and the expected contribution it will make to the medical community.
5. The introductory section of this paper explains that ALL cure rates are already relatively favorable in the existing context, so what is the significance of conducting this paper? Do the findings of this paper further mitigate the adverse effects of ALL? Please provide an explanatory note on this.
6. The interaction relationship between immune cells is an important factor leading to ALL, and thus it is suggested that the introductory section should elaborate on what mechanism of interaction exists between immune cells and what role this mechanism plays in revealing the relevant details of the progression of ALL.
7. The description of the results in Figure 2 is not clear enough, and it is suggested to state what phenomenon was used in this paper to clarify that B-cells are an important factor in the pathogenesis of ALL.
8. The description of the results in Figure 4 seems to be somewhat meaningless, and since existing studies have shown that MYC and STAT1 are already TFs that regulate cell proliferation, what is the significance of continuing to study these two TFs in the later section? Please revise the description.
9. The paragraphing between lines 280-282 is incoherent, please revise this and review the issue throughout the text to avoid non-compliance with publication requirements.
10. This paper reveals the existence of interactions between B cells and other cells, which may be one of the elements influencing the progression of ALL, and thus suggests that a few typical ligand-receptor pairs should be selected for a comprehensive description and added to in the discussion section, thus making the paper more complete.

Experimental design

no comment

Validity of the findings

no comment

---

## Round 0.2 · accepted · Accept

Both reviewers have recommended acceptance based on your revised submission. I have reviewed the changes and can confirm that you have adequately addressed all of the reviewers' comments. The manuscript will now proceed to the next stage of the publication process. You will receive further instructions from the editorial office shortly.

Reviewer 1 ·

Basic reporting

In this study, the authors explored the potential association between the stemness of specific B cell clusters and the development and prognosis of acute lymphoblastic leukemia. The prognostic factors and mutation characteristics of acute lymphoblastic leukemia were identified through bioinformatics methods and validated in vitro experiments. I have carefully read the revised manuscript by the author, and the quality has been significantly improved. They have also responded carefully to the reviewer's comments, basically solving the reviewer's doubts. I no longer have any new comments.

Experimental design

no comment

Validity of the findings

no comment

Reviewer 2 ·

Basic reporting

no comment

Experimental design

no comment

Validity of the findings

no comment

Additional comments

In this study, sample data of ALL and aggregated cell subsets were obtained from public databases by single cell analysis. The relationship between immune cell subsets and disease progression, metabolic regulation and immune regulation in ALL was further explored and verified by cell experiments. This is a representative dry-wet combination study, using bioinformatics to mine molecular markers and using experiments to verify that the main purpose of this study is to study the infiltration types of immune cell subsets in acute lymphoblastic leukemia (ALL) by single cell transcriptome analysis. The author replied to my comments in detail, and I think the overall requirements for publication have been met.